# Disparities in pre-health advising across California's public universities

**Francine Rios-Fetchko**[1]*, **Mariam Carson**[1], **Manuel Tapia**[1], **Alicia Fernandez**[1‡],
**Janet Coffman**[2‡]

**1** Latinx Center of Excellence, University of California, San Francisco, San Francisco, California, United States of America, **2** Institute for Health Policy Studies, University of California, San Francisco, San Francisco, California, United States of America

‡ AF and JC are joint senior authors on this work.
* francine.rios-fetchko@ucsf.edu

## Abstract

### Background

The Supreme Court's decision in Students for Fair Admissions, Inc., v. Harvard College is likely to result in the matriculation of fewer students from historically excluded racial/ethnic groups at more selective colleges and universities and matriculation of more students at less selective colleges and universities. Because of this, it is important to understand how resources for pre-health advising, a modifiable factor that can help increase the diversity of the health workforce, vary across institutions with differing levels of selectivity. Colleges are known to vary in resources, structure, and investment in pre-health advising but data are lacking and there is no estimate of any pre-health advising resource gap.

### Purpose

To quantify availability of advising resources and identify perceived challenges in pre-health advising in California's highly diverse public and select private undergraduate institutions.

### Methods

Structured 60-minute Zoom interviews from June 2022 –October 2022 at 18/23 CSU (California State Universities), 9/9 University of California (UC) institutions and 6 select private institutions with varying levels of selectivity. Two investigators independently analyzed interviews using a Grounded Theory Approach. The full study team reviewed transcripts and themes.

### Key results

Pre-health advisor capacity varied greatly across the three types of institutions. CSU: mean = 1 FTE advisor: 24,620 graduates (range: 1: 1,059–1: 150,520); UC mean = 1 FTE advisor: 4,526 graduates (range: 1: 1,912–1: 10,920); private institutions mean = 1 FTE advisor:1,794 graduates (range: 1: 722–1: 5,300). Participants reported common challenges: advising capacity, lack of advisor training, advisor turnover, and student difficulties in accessing clinical opportunities and required coursework. CSU and UC participants noted

**Data Availability Statement:** All relevant data are within the manuscript and its Supporting information files. In order to protect participant privacy, transcripts from this study will not be made publicly available. Deidentified quotes from

participant interviews have been included in the manuscript, all other data sharing would violate participant consent agreements approved by the University of California, San Francisco Institutional Review Board (Study # IRB # 21-34656). Requests to review additional transcript excerpts should be directed to the UCSF IRB, reachable at (415) 476-1814.

**Funding:** AF, JC received funding from the California Health Care Foundation Grant #A138605. https://www.chcf.org/ AF and JC contributed to study design, data analysis, decision to publish and preparation of the manuscript.

**Competing interests:** The authors have declared that no competing interests exist.

that these had greatest impact for first generation and racially/ethnically underrepresented students for whom lack of informal professional networks, lack of other mentors, and financial responsibilities complicate college navigation and professional school application.

## Conclusions

Students at CSU campuses had 5 times less access to pre-health advising per graduate than UC students, and 13 times less than students at private institutions. Much greater investment is needed in California's public institutions, particularly CSUs, to increase equity in access to advising for pre-health professional students. Research should examine pre-health advising resource capacity in other states, especially those that are now facing race-neutral admissions policies at undergraduate institutions and health professions schools.

## Introduction

The SCOTUS decision in Students for Fair Admissions vs Harvard College [1] does away with affirmative action in admissions to colleges and universities, changing the educational landscape in much of the country, although not in California, which abolished consideration of race in admissions in 1996 [2]. The resultant 27 years of experience in California strongly suggests that the SCOTUS decision will result in fewer students from historically excluded racial/ethnic groups matriculating at more selective institutions [3], and a greater concentration of these students at less selective institutions. A recent analysis found that only 6% of U.S. college students attend an institution with an acceptance rate of less than 25% [4, 5]. Requiring race neutral admissions policies is likely to decrease the number and percentage of students from historically excluded racial/ethnic groups attending selective institutions, which have historically been the primary sources of medical school applicants. Because of this, it is important to understand differences across institutions in resources available to historically excluded students who aspire to careers in medicine and the health professions. The experience in California may be informative to states now facing race-neutral admissions policies at undergraduate institutions and health professions schools.

Prior research has found an association between pre-med advising and medical school application and matriculation numbers [6, 7]. The relationship between advising and application is especially important for historically excluded students, who may drop off the educational medical pathway at higher rates than their counterparts from racial/ethnic groups that have not been historically excluded [7–9]. Mentorship and advising can provide critical support for prospective applicants, especially for those without informal guidance from family members or friends in the health professions.

Increasing the numbers of medical students from historically excluded racial/ethnic groups is important because racial/ethnic concordance has been found to increase patient trust and improve many health outcomes. In 2022 Hispanics/Latinos and Black/African Americans represented 12% and 10% of medical school matriculants, despite making up 19% and 14% of the US population, respectively [10, 11]. There is less diversity among current practicing physicians [12]. In California, where Latinx make up 39% of the population and only 6% of the physician workforce, the imperative to increase the diversity of the workforce is even more salient [13].

California's public university systems—the California State University System (CSU) and the University of California System (UC)—are a potential source of diverse medical students

because they have large numbers of graduates from racial/ethnic groups that have been historically excluded from medicine. In addition, both systems have been lauded as agents of social mobility and both have leaders who advocate for diversity, equity, and inclusion efforts [14–16]. However, despite producing approximately 175,000 graduates per year [17], the CSU and UC campuses have a relatively poor record of producing applicants and matriculants to medical school. In 2019, across the 23 CSU campuses (107,000 bachelor's degrees awarded [18]) there were only 273 total applicants to medical school, of which 78 were Latinx and none were African American [19]. The 9 UC campuses (62,906 bachelor's degrees awarded [20]) produced 426 Latinx and 161 African American applicants to medical school. In contrast, Stanford University, with 1,893 graduates, had 40 Latinx students apply to medical school that the same year and alone accounts for about 8% of California's Latinx medical school matriculants.

While family socio-economic background [21] and many other factors contribute to the numbers and characteristics of students applying to medical school, undergraduate educational climate and support for students in pre-health careers are crucial modifiable factors. Institutions such as Xavier University, whose graduates comprise a disproportionally sizable percentage of Black medical students, have robust pre-medical and pre-health advising programs [19]. Recognizing the important role of pre-medical advisors, we set out to understand pre-medical/pre-health profession advising capacity, strengths, and challenges across the CSU and UC systems.

We conducted interviews with pre-health advisors at the CSUs, UCs, and select California private institutions with varying levels of selectivity to determine availability of pre-health profession advising across campuses, and to ascertain the perspectives of pre-health advisors on their work. Specifically, we sought to 1) estimate access to advising for CSU and UC students relative to one another and to students at selected private institutions in California, and 2) understand advisor perspectives on challenges to serving pre-health students at the state's public universities.

## Methods

We recruited pre-health advisors between May 1st, 2022 and October 1st, 2022 using publicly available data to identify the highest-ranking advisor, or if none was found, a university official. Two research analysts communicated with advisors or officials at the 22 CSU campuses (excluding one specialized college), the 9 UC campuses that enroll undergraduates, and 6 selected private institutions in California. The six private institutions were chosen to represent a range of admission selectivity corresponding to the range exhibited in the public institutions from 6% to 75% (Table 1) [22]. Interviews were conducted between June and October 2022. This research was approved by the UCSF Institutional Review Board (IRB # 21–34656).

**Table 1. Size, location and acceptance rate of six selected private institutions.**

|   | Undergraduate Numbers | Location | Acceptance Rate |
|---|---|---|---|
| 1 | 5–10k | Northern California | < 5% |
| 2 | <5k | Southern California | 10–15% |
| 3 | 20–25k | Southern California | 10–15% |
| 4 | <5k | Southern California | 50–55% |
| 5 | 5–10k | Northern California | 50–55% |
| 6 | 5–10k | Northern California | 70–75% |

### Interviews

Participants received a survey on advisor FTE in advance of a semi-structured interview. The interview explored advisors' perceptions of the strengths, weaknesses, and quality of pre-health advising at their institutions. All interviews were conducted over Zoom for 60–90 minutes and verbal consent to participate and be recorded was obtained. One interviewee refused recording. Recorded interviews were professionally transcribed; de-identified recordings and transcripts were secured.

### Analysis

Interviews were analyzed using an inductive approach, consistent with modified Grounded Theory approaches [23]. Two research analysts read a random sample of 8 transcripts and generated a preliminary codebook using Jamboard, a tool for visualizing and grouping recurring themes. This codebook was iteratively refined through discussions with the entire research team and applied to the remaining transcripts using Dedoose software to facilitate line-by-line coding. Themes were finalized through iterative discussions among the entire research team.

Publicly available data regarding numbers of graduates were used to estimate the ratio of FTE pre-health advisors to graduates. Pre-health advisor FTEs were obtained from interview participants and subsequently confirmed over email with each participant to ensure accuracy.

## Results

We interviewed 33 pre-health advisors (response rate = 89%): 18 from CSU, 9 from UC, and 6 from private institutions (See S1 Table for details). We have grouped responses into the following areas: access to pre-health advising, structure of pre-health advising, advisor perspectives on structural challenges, individual advisor challenges, and advisor perspectives on student challenges.

### Access to pre-health advising

Ratios of graduates per FTE pre-health advisor varied across CSU, UC, and private institutions (Table 2): CSU mean = 1: 24,620 graduates (range: 1: 1,059–1: 150,520), UC mean = 1: 4,526 graduates (range: 1: 1,912–1: 10,920) and private institutions mean = 1:1,794 graduates (range: 1: 722–1: 5,300) (Fig 1). On average, CSU campuses had 5.4 times more graduates per advisor than UC campuses, and 13.7 times more graduates per advisor than private institutions (Table 2). Large differences also existed within each of the two public systems. The largest variation was found across CSU campuses, which ranged from 1,000 to 150,000 graduates per FTE advisor. (Additional data is presented in S1 Table located in the supporting information section).

### Structure of pre-health advising

Participants described five distinct types of pre-health advising structures: 1) no pre-health advising, 2) pre-health advising offered by a faculty member, 3) pre-health advising as part of general academic advising, 4) pre-health advising as part of career center advising, and 5) an independent pre-health advising office (Table 2).

Three CSU participants confirmed that their campuses have no pre-health advisors; no UCs or privates reported having no pre-health advisors. In addition, 7 of the 8 institutions where the pre-health advising structure consisted of a faculty member offering pre-health advising in addition to teaching their regular courses were CSU campuses. Only 11% (2/18) of

**Table 2. FTE and office structure of pre-health advising in California universities.**

|  | CSU (n = 18) | UCs (n = 9) | Selected Privates (n = 6) |
|---|---|---|---|
| **Mean Grad/FTE (SD)** | 24620 (36008) | 4526 (2971) | 1749 (1778) |
| **# of Grads /FTE** |  |  |  |
| No Pre-health Advising | 3 (18%) | 0 (0%) | 0 (0%) |
| Grad/FTE > 20,000 | 8 (44%) | 0 (0%) | 0 (0%) |
| Grad/FTE 10,000–20,000 | 1 (6%) | 1 (11%) | 0 (0%) |
| Grad/FTE 1,000–10,000 | 6 (33%) | 8 (89%) | 3 (50%) |
| Grad/FTE <1,000 | 0 (0%) | 0 (0%) | 3 (50%) |
| **Office Structure** |  |  |  |
| None/no FTE | 3 (17%) | 0 (0%) | 0 (0%) |
| Professor with advising responsibilities | 7 (39%) | 0 (0%) | 1 (17%) |
| General Academic Advising | 4 (22%) | 2 (22%) | 1 (17%) |
| Career Center | 2 (11%) | 4 (44%) | 1 (17%) |
| Dedicated Pre-Health Office | 2 (11%) | 3 (33%) | 3 (50%) |

FTE–Full time effort; SD–Standard deviation; CSU–Cal State University campuses; UCs–University of California campuses

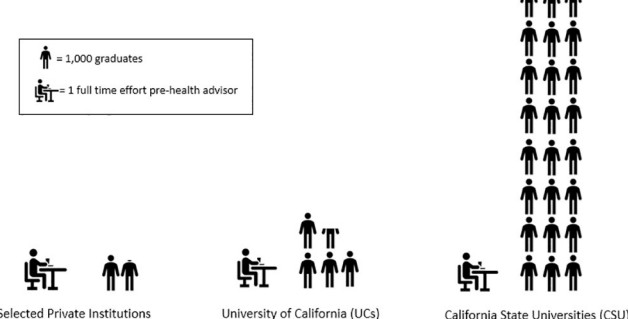

**Fig 1. Graduates per full time effort pre-health advisor, California higher Ed institutions.**

CSU campuses had independent pre-health advising offices, compared to 33% (3/9) of UC campuses and 50% (3/6) of private institutions.

Advisors in independent pre-health advising offices were largely content with their office structure in part due to their ability to focus exclusively on pre-health advising and, for many, their budgetary autonomy. One advisor explained that:

> I think it gives us the autonomy. I guess the best way to explain it is what it looks like not to be a separate unit. When I did this similar job in another university, 60% of my job was pre-health advising. As a specialist in that area, I was in a career center; 40% of my job was everything else. My time was split. Same number of students. The university was comparable in size, but I had multiple jobs that I had to do, and I didn't have the autonomy to do just pre-health advising. Being housed separately, we oversee ourselves. As the director, I have my own budget. I run it. I determine peer staffing. I determine where we can save–how we're spending our funds, where we're spending our funds, and that leads us to being able to be more robust,

*and again I'm just very focused. This is all I do every day. I'm not torn in non-pre-health directions.*

*- UC campus advisor*

In contrast, pre-health advisors who doubled as faculty members cited the most frustration with their 'office' structure, pointing toward the difficulty of managing multiple roles. One advisor explained:

*The way that we do it is very informal and there's no rules–it honestly it comes down to the advisor and the quality of the advisor and the interest that the advisor has, and I think I would love to see that change. I'm not saying I'm doing a bad job. It just has to fall down on the priority list because I have other stuff to do as I have all these classes to teach and grants to write and stuff like that. Then of course this is going to fall by the wayside a little bit. Yes, to make it more formal at my school would make it I think a lot more useful for students. Especially students that are underrepresented.*

*- CSU campus advisor*

Advisors housed in both general academic advising and career centers noted challenges around visibility and access to specific resources, such as test prep materials, while also highlighting the ability to offer multiple services in one place as an advantage of their institution's pre-health advising structure.

## Advisor perspective on structural challenges

Many advisors perceived limited institutional investment in advising, competing priorities for advising, and lack of coordination between community colleges and bachelor's degree granting institutions.

**Insufficient institutional investment in advising.** Advisors from across CSU, UC and private institutions reflected on challenges perceived as arising from insufficient investment in pre-health advising. Advisors reported not feeling sufficiently appreciated or paid, which they perceived as fueling advisor burnout:

*I'm going to be honest. I don't believe that the administration puts enough value and resources into pre-health advising, either because they have bigger fish to fry and bigger problems, or because they don't think it's important.*

*- CSU campus advisor*

One advisor from a private institution described his institution as "a business," and explained that "unless you can convince the administration that if you don't put some money in the pre-health advising, you're actually going to dimmish the way you look to your customer's base. I don't think they're going to do it."

**Competing priorities for advising.** In many cases, advisors cited administrative turnover and shifting of college priorities as a driver of confusion and ineffective advising. Many CSU advisors pointed toward the tension between advising to fulfill institutional priorities regarding graduation rates within a given enrollment time and advising students who seek to matriculate at a health professions school who may benefit from a slower advancement rate:

*If we ignore kids' desire to go into pre-health programs, the university and higher education in general—how do you judge a program's success? Graduation rate and retention rate. Especially in the CSU system, there is a large focus on graduating in four years or getting as many kids through the system, but the reality is that I think many of our CSU kids, especially here, they were carrying part-time jobs. So, you start running into this almost conflict of interest where, okay, is your priority to get them to graduate or is the priority to maybe have them slow down a bit, do a little bit better in their classes and make them more competitive applicants? So, there's a little bit of mixed messaging here where would you rather have a student graduate in four years and not really have a good academic record or would you rather have them stick around for five years and do better in their classes because they've had to work part-time jobs? That's not something that I feel like anyone has really addressed because the message is always "We need these kids to graduate in four years."*

*- CSU campus advisor*

Advisors at some UCs also noted institutional pressures to prioritize fulfillment of undergraduate major requirements over other interests, although such comments were less common. No private institutions noted institutional pressure related to increasing graduation rates.

**Lack of coordination between community colleges and bachelor's degree-granting institutions.** Advisors also reported a lack of coordination between community colleges and their institutions that can hamper students on their path to health professional school matriculation. Students may lack pre-health advising at the community college level and be mistakenly enrolled in classes that do not count toward either their major (at CSU or UC) or toward health professional school matriculation, often costing them extra time and money to complete additional classes [24]. This issue most saliently affects the CSU campuses, who have a greater proportion of community college transfers than the UCs and private institutions [25].

## Individual advisor challenges

Interviewees across all three types of institutions highlighted challenges that included limited advisor capacity, high advisor turnover, and lack of training.

**Advisor capacity.** Nearly all advisors reported wanting to do more for their students but lacking the time to do so. One CSU advisor summed it up as, *"honestly just simply trying to stay afloat."* Advisors explained that lack of time was often what prevented them from doing things such as data collection, targeted outreach, in-depth advising (including essay editing and application review) or keeping current with pre-health admission requirements. Advisors who serve large populations of students, lack team members to support them, or have additional responsibilities such as teaching, research, or general/career advising reported being most constrained by capacity. These challenges are most common at CSU campuses and, to a lesser extent, UC campuses. One CSU advisor stated:

*I'm going to be honest, we don't have any strengths [in our office]. We don't have vision. We don't have resources. It's very vague. They hired me and they saw I had health professional advising background and said, "Hey, you're our new health professional advisor," and that's it. For me, it's just me and I'm not dedicated to pre-health. I'm general advising and somehow I got to figure out a way with my other responsibilities to [incorporate] professional advising. So we don't really have a structure but we need one.*

*- CSU campus* advisor

One UC advisor reflected on how limited advisor capacity impacted students' ability to access services explaining that their *office "gets scheduled for three weeks out within 24 hours"* of opening an appointment. Another UC advisor compared their student load to that of their private university counterparts:

*I think another challenge for us, honestly, is being at a large university, there's so many students that are pre-health. It's funny talking to some colleagues at private institutions, they're like, "Oh yes, we had a group of about 20 pre-med students this year," and I'm like, "What?"*

*–UC campus advisor*

### Advisor turnover and training

Interviewees, particularly at CSUs, noted high rates of advisor turnover that often left positions unfilled for long periods of time, posing a challenge to developing clear strategies, building relationships with students, and developing expertise on the job. One advisor reported that their only pre-health advising position was unfilled for over a year, while another noted that at least one of their advisors was out on leave for each of the past 3 years. A third advisor noted that those leaving were often advisors from underrepresented racial/ethnic groups and pointed toward low pay and lack of support from the institution as driving factors for turnover:

*We're watching a mass exodus of people of color say, "There are other places that want me and will pay me for my time and my energy," and I think that that's a really big deal. I mean if they're not feeling passionate and supported, like I think they're passionate about working with our students but they're not feeling the love from the administration.*

*- CSU campus advisor*

In contrast, one advisor at a private university highlighted their office's ability to retain staff as a strength of their advising system. They explained:

*We have quite a large staff. We have a group where many of our staff have been with the office since its inception 11 years ago. I think we're really able to come together and find creative ways to support the students. I think that the staffing that we have allows us to do that. We also have a decent budget.*

*–private school advisor*

Advisors from CSU campuses and those who were faculty members or worked in general academic/career advising offices were most likely to note that they had no formal training upon entering their position. One advisor explained the challenges of not having any training upon hiring:

*Advisors are thrown into this role without training. We train each other the best way possible, but we don't have an official training process. When I first started, I was handed a catalog and I was told to read it. That was my training and then you really can't learn everything out of the catalog because every situation, every student is so different. Their needs are so different.*

*- CSU campus advisor*

Some advisors expressed concerns that other advisors or faculty members were not up to date on admissions requirements and may be sharing incorrect, and possibly detrimental, information with their students. As one CSU advisor explained, "*. . .a lot of advisors, they're handing out information that's antiquated and it's not current.*"

On the other hand, schools with independent, career center or general academic advising structures less often cited lack of training as a challenge, with some even pointing toward their expertise in the field as an asset for students.

## Advisor perspectives on student challenges

Many advisors also noted challenges that students from backgrounds historically underrepresented in medicine (UIM) commonly faced, including lack of access to clinical volunteer opportunities, timely early advising, and multiple compounding barriers related to economic and familial pressures. Students considered to be UIM include those who identify as: Black/African American, Hispanic/Latino, Native American (American Indians, Alaska Natives, and Native Hawaiians), Pacific Islander, and mainland Puerto Rican as well as students from economically or educationally disadvantaged backgrounds [26].

**Student access to clinical experiences.** Advisors noted that gaining clinical experience was substantially more difficult for students from underrepresented backgrounds who may not have access to health professionals in their social networks. Advisors working in colleges in rural areas of the state also noted difficulty connecting their students with direct clinical work. While advisors noted that gaining clinical experience became much more difficult for everyone with COVID, they also noted disproportionate challenges for UIM students. As one advisor explained,

> *The major hurdle that students say that holds them back is gaining real-world experience: shadowing. Especially when you're dealing with populations of minorities or [first-generation students], they don't have connections, right? They don't have family members that are physicians. They don't have close acquaintances that they could "Go work with so-and-so over the summer," and they're really at a loss in terms of how to gain experience because as you could imagine, 99% of the people that you just send an email to and say, "I'm looking to shadow," gets no response.*
>
> *- CSU campus advisor*

**Timely access to advising.** Nearly every advisor expressed wanting a better system for identifying students as pre-health early on in their college careers, as early identification can ensure students have information regarding admission requirements, availability and timing of required courses, and access to academic supports. Advisors perceived this as especially pertinent for students who transfer from community colleges or are the first in their families to navigate the college landscape.

**UIM students face complex and disproportionate challenges.** While all participants pointed to challenges faced by all pre-health students such as access to clinical activities and challenging academic coursework, advisors serving students from underrepresented backgrounds perceived additional challenges for their students. These included: financial pressures, having to work while in college, familial responsibilities, lack of access to academic resources before college, feelings of imposter syndrome, being the first in their family to navigate college, and language barriers. Advisors from all three university systems noted that UIM students at

their campuses face disproportionate challenges, however, discussions of such challenges were more salient at the CSUs due to the high number of UIM students they serve.

As one advisor observed:

*The starting point in education is different, right? So, kind of related to finance, but also socioeconomic, geographic factors that play into it, where the people might not necessarily be getting access to the same level of education and support. Primarily, support within learning, learning styles, and being able to handle it, especially since I work with a lot of first-generation students. It's their first time tackling a lot of challenges, so that makes it even more difficult because they don't have a reference point.*

*- UC campus advisor*

One advisor suggested that '*intrusive advising*'—going beyond academics and addressing challenges affecting all aspects of their students' lives—was necessary to serve underrepresented, first-generation students effectively and appropriately. Furthermore, many advisors highlighted the non-traditional routes many of their students take to enter the health-professions, emphasizing the inadequacy of a "one-size-fits-all" approach to advising that does not consider students' unique circumstances and the corresponding need for tailored 1-on-1 advising.

## Innovations in advising

To address limited capacity and resources, some pre-health advisors and advising offices adopted innovations to increase their visibility, improve office efficiency, and augment the opportunities available to their students (Table 3).

Two examples, implemented across multiple campuses, stood out as particularly beneficial to students: (1) exploratory classes for different health professions, and (2) employing advisors with backgrounds reflective of the student body. Several institutions had developed a pre-health exploratory class, often offered for credit, and taught by advisors themselves. These classes were praised for exposing students to a variety of different health professions while enabling them to develop relationships with advisors early in their undergraduate careers. Multiple advisors reported that students benefit from having advisors who share their background. In particular, advisors from historically disadvantaged backgrounds found that having shared experiences with their advisees made it easier for them to connect with UIM students, understand the various challenges they face, develop trusting relationships and serve as effective advisors.

Finally, advisors from across the CSU system have created a network to share experiences, challenges, and best practices and begin to standardize advising efforts across the CSU system.

## Discussion

In this study of pre-health advising in California we found that advisors reported major differences in availability of pre-health advising for students at CSU, UC, and selected private institutions. Students at CSU campuses had 5 times less access to pre-health advising per graduate than UC students, and 13 times less than students at private institutions with comparable levels of selectivity. These data may be informative for other states that are now facing race-neutral admissions policies at undergraduate institutions and health professions schools, which are likely to result in an increased concentration of students from historically excluded racial/ethnic groups at less selective institutions, such as public university systems.

**Table 3. Key innovations in pre-health advising in California universities.**

| Key Innovation | Description |
|---|---|
| Committee Letter | A committee letter is a letter from multiple faculty and advisors in support of a student's application to medical school. To receive a committee letter of support a student must apply and interview with the letter writing committee, allowing faculty and staff to take a close look at their application before official submission to medical school. Some advisors report increased acceptance rates for students with committee letters due to the high-touch advising it requires. However, committee letters may also present an additional obstacle for students are not selected and therefore do not receive institutional support when applying. |
| Peer Advisors/Student Workers | Many colleges have leveraged undergraduate employees or volunteers to serve as near-peer advisors and supply general pre-health advice, giving the professional advisors time for more individually tailored advising appointments. Although peer advisors can be a good resource, they should not take the place of professional advisors and should be supervised closely due to their lack of experience and training in the field. |
| Health Professions Career Exploration Course | Exposing first- or second-year undergraduate students to different health professions such as medicine, nursing, physician assistant, physical therapy, etc. can help students make educated decisions about their future. |
| Advisors with Shared Experiences | Advisors who are culturally and linguistically representative of the students they serve have found success in being able to relate to the lived experiences of their advisees. This allows for tailored advising and in-depth relationship building. |
| Connections with local physicians | Advisors have developed shadowing programs in partnership with local clinics and physicians, allowing students to gain clinical exposure prior to applying to health professional schools. Advisors noted that shadowing opportunities have been limited since COVID-19 restrictions were put in place. |
| Pre-health conference | One UC campus has developed a large, yearly pre-health conference that was cited by advisors across the state as a source of important information exchange and networking for both students and advisors. |
| Enhanced Academic Success Experience | Students who take part in the enhanced academic success experience receive additional tutoring and support in their classes, setting them up for success in science coursework. |
| Summer Bootcamp | A summer bootcamp designed by one college helped students prepare for internships and other health career focused opportunities by going over applications, assisting with essay writing, and providing interview preparation in an intensive, high-touch, program. |
| Student Club Partnerships | Partnerships between pre-health advisors and student pre-health clubs help advisors reach increased numbers of students. Advisors noted that rapid turnover of club leadership poses a challenge to continuity of partnerships. |
| Connections with Community College | Some advisors have created relationships with community college advisors, sharing information and tools with them to promote the success of transfer students looking to enter the health professions. |
| Advisor Networks | Communicating with different pre-health advisors through formal professional organizations, or informal networks, both within and across college systems creates a sense of unified mission and allows advisors to share experiences and resources. |

In addition to large disparities in access to advising across CSU, UC, and private institutions, we found high variability is access to pre-health advising resources within the CSU and UC systems. Advisor burden at the CSUs ranged from around 1,000 graduates per advisor to over 150,000 graduates per advisor. Furthermore, many advisors at both CSUs and UCs reported a lack of any formal training for their job, and CSU advisors felt burdened by what they perceived as a lack of alignment between institutional prioritization of four-year graduation rates and what they felt necessary to best prepare students for professional schools.

Advisors at the CSUs and UCs also reported many challenges in addressing the needs of historically excluded and first-generation students who face multiple barriers to medical

school admission. Due to capacity constraints, they often cannot provide the individualized and intensive advising they perceive these students need to help address competing priorities and barriers. In a cruel parallel of the "inverse health care law," paradoxically those with more need for pre-health advising are advised by people with less time and training.

Our study has several limitations. Our estimate of FTE is inexact, particularly when advisors reported multiple roles and responsibilities. Second, we only selected 6 of the many private institutions and universities in the state to include in our analysis. Although we selected private institutions with a range of selectivity, size and geographic location to capture a range of possible responses, we made no attempt to capture the full range of advising resources offered by private institutions. Finally, four CSU schools did not respond to our request to interview.

Educating the most ethnically and economically diverse student population in the nation, California's public universities can serve as a national model of programs and resources that increase the diversity of the health workforce. However, without the structures and resources in place to successfully usher students to the next step in their professional careers, these institutions cannot fully fulfil their educational mission. Our results suggest that the CSU and UC systems should (1) establish pre-health advisor to student ratios and fund additional advisors where needed to ensure that students at all campuses have equal access to pre-health advisors; (2) establish training standards for all pre-health advisors; (3) recognize the importance of holistic, individualized advising for students from underrepresented, first generation, or non-traditional backgrounds; (4) consider establishing an independent pre-health office at each campus or for multiple campuses in a region; and (5) signal institutional support for pre-health advising by providing competitive compensation to pre-health advisors. Local healthcare systems and physicians could also step up to provide additional clinical experiences and paid internships for students during college. Implementing these recommendations would enhance the ability of California's public universities to serve as engines of social mobility and address the state's need for a racially and linguistically diverse physician workforce.

## Supporting information

**S1 Table. Additional campus data.**
(DOCX)

## Acknowledgments

Contributors: We would like to thank the participating pre-health advisors for their time and contribution to this research.

Previous Presentations:

Results from this study have been presented at the following conferences:

Western Association of Advisors for the Health Professions Annual Conference. May 2023.

Society of General Internal Medicine Annual Conference. May 2023.

Association of American Medical Colleges Health Workforce Research Conference. May 2023.

## Author Contributions

**Conceptualization:** Francine Rios-Fetchko, Mariam Carson, Manuel Tapia, Alicia Fernandez, Janet Coffman.

**Data curation:** Francine Rios-Fetchko, Mariam Carson, Manuel Tapia, Alicia Fernandez, Janet Coffman.

**Formal analysis:** Francine Rios-Fetchko, Mariam Carson, Manuel Tapia, Alicia Fernandez, Janet Coffman.

**Funding acquisition:** Alicia Fernandez, Janet Coffman.

**Investigation:** Manuel Tapia.

**Methodology:** Manuel Tapia.

**Project administration:** Francine Rios-Fetchko.

**Writing – original draft:** Francine Rios-Fetchko, Mariam Carson.

**Writing – review & editing:** Francine Rios-Fetchko, Mariam Carson, Manuel Tapia, Alicia Fernandez, Janet Coffman.

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
