## [Decision Letter · Decision Letter 0]

23 Oct 2023

PONE-D-23-24270Disparities in Pre-Health Advising Across California’s Public UniversitiesPLOS ONE

Dear Dr. Rios-Fetchko,

Thank you for submitting your manuscript to PLOS ONE. After careful consideration, we feel that it has merit but does not fully meet PLOS ONE’s publication criteria as it currently stands. Therefore, we invite you to submit a revised version of the manuscript that addresses the points raised during the review process.

It is agreed that the article should be accepted with minor revisions. The revisions are as follows:More details regarding the use of grounded theory are warranted (page 6, lines 145 – 153).  Describe grounded theory and what was done.  Provide the processes involved in conducting this analysis.  For example, Reviewer 1 questioned “whether the authors really do line-by-line coding with no presumptions of what they might find.”  What strategies were used in analyzing the data (e.g., deductive and/or inductive strategies; abductive analysis such as Tavory and Timmermans, 2014). More clarity on the analytic technique and citation to the closest methodological approach would strengthen the article.Define the term UIM before using it. It is first mentioned on page 14, line 308.   Indicate what the letters stand for and who is included.Given that data from 3 different types of university systems are compared (i.e., CSU vs UC, vs private schools), the reviewers recommend that for each of the identified themes, examples from each of the university systems be included.  This allows the reader to have a greater understanding of institutional variability or commonality.   For example, for the theme “Competing Priorities for Advising” a reviewer noted that “there was not a discussion of institutional variation. And the quote provided from a CSU campus advisor is a phenomenon that may occur more frequently at public schools with first-generation and low-income students. “ It is agreed that Table 3 provides an excellent example of potential innovations in pre-health advising.  Reviewer 1 cautions it may be useful to note potential limitations for some of the recommendations.  For example, peer advisors who serve as near-peer advisors sitting in for trained staff may provide inadequate or incorrect information to pre-health advisees. It is useful to note in this table, as was done for the “Committee letter”.

Overall, the topic is very interesting and has the potential to address factors associated with advising and racial/ethnic disparities in the health workforce. You are commended for your attempt to add to the extant literature on the topic of advising pre-health students.  While the study focuses exclusively on pre-health students in California, it has implications for advising for all students attending institutions of higher education.  

We look forward to receiving your revised manuscript.

Kind regards,

Jocelyn Octavia Turner-Musa

Academic Editor

PLOS ONE

“AF, JC received funding from the California Health Care Foundation Grant #A138605.

https://www.chcf.org/

AF and JC contributed to study design, data analysis, decision to publish and preparation of the manuscript.”

“Funding: This research is partially funded by The California Healthcare Foundation (A138605).”

“AF, JC received funding from the California Health Care Foundation Grant #A138605.

https://www.chcf.org/

AF and JC contributed to study design, data analysis, decision to publish and preparation of the manuscript.”

6. We notice that your supplementary table (Appendix A) is included in the manuscript file. Please remove them and upload them with the file type 'Supporting Information'. Please ensure that each Supporting Information file has a legend listed in the manuscript after the references list.

Reviewers' comments:

Reviewer's Responses to Questions

**Comments to the Author**

1. Is the manuscript technically sound, and do the data support the conclusions?

Reviewer #1: Yes

2. Has the statistical analysis been performed appropriately and rigorously? 

Reviewer #1: N/A

3. Have the authors made all data underlying the findings in their manuscript fully available?

Reviewer #1: Yes

4. Is the manuscript presented in an intelligible fashion and written in standard English?

Reviewer #1: Yes

5. Review Comments to the Author

Reviewer #1: Review for “Disparities in Pre-Health Advising Across California’s Public Universities”

PONE-D-23-24270

This study provides a concrete review of pre-health advising disparities in the state of California. CA is a useful case study because the implementation of Prop 209 in 1996 effectively did away from affirmative action for racially marginalized students. In the wake of the Supreme Court decision in Students for Fair Admissions, Inc. v. Harvard College, there are strong reasons to anticipate that pre-health students of color around the country will attend less selective institutions than they might have otherwise. The study attempts to gauge what kind of resources these students will find at less selective institutions. Resources matter for successful movement into health disciplines. Indeed, we see in CA huge differences between the racial background of state residents and representation of Latinx health providers.

I am grateful to see clear information and comparisons across sectors in the state and find this article to be quite helpful in documenting disparities. The framing for why the problem is worthy of study is effective.

I have a few suggestions to improve the piece.

1. Define UIM before using the term. Indicate what the letters stand for and who is included.

2. Scholars frequently say they are doing grounded theory when they are not. Did the authors really do line-by-line coding with no presumptions of what they might find? My guess is that this article utilized more movement between deductive and inductive strategies (abductive analysis, Tavory and Timmermans 2014). Please provide a bit more clarity on the analysis technique and a citation to the closest methodological approach.

3. Within each point, the authors need to give a sense of how common this was at CSU, vs. UC, vs. privates. They often do this well. However, on the “Competing Priorities for Advising” there was not a discussion of institutional variation. And the particular quote provided from a CSU campus advisor is a phenomenon that occurs, I would assume, much more frequently at public schools with first-gen and low-income students. Please contextualize.

4. I love the table of potential innovations but note caution with peer advisors. I have also see cases where peer advisors sitting in for trained staff provide inadequate or incorrect information and are a poor band-aid for limited pre-health advising. It is useful to note in this table potential limitations of some of these remedies (as is done for the committee letter).

6. PLOS authors have the option to publish the peer review history of their article (what does this mean?). If published, this will include your full peer review and any attached files.

Reviewer #1: No

---

## [Author Response · Author response to Decision Letter 0]

5 Dec 2023

Please see below for our responses to each of the editor and reviewer's comments. This information may also be found in a table format in the Response to Reviewers attachment. Thank you very much for your time and consideration.

Editor 

1. More details regarding the use of grounded theory are warranted (page 6, lines 145 – 153). Describe grounded theory and what was done. Provide the processes involved in conducting this analysis. For example, Reviewer 1 questioned “whether the authors really do line-by-line coding with no presumptions of what they might find.” What strategies were used in analyzing the data (e.g., deductive and/or inductive strategies; abductive analysis such as Tavory and Timmermans, 2014). More clarity on the analytic technique and citation to the closest methodological approach would strengthen the article. 

Thank you for this comment. Lines 146-154 now read:

Interviews were analyzed using an inductive approach, consistent with modified Grounded Theory approaches. Two research analysts read a random sample of 8 transcripts and generated a preliminary codebook using Jamboard, a tool for visualizing and grouping recurring themes. This codebook was iteratively refined through discussions with the entire research team and applied to the remaining transcripts using Dedoose software to facilitate line-by-line coding. Themes were finalized through iterative discussions among the entire research team. 

2. Define the term UIM before using it. It is first mentioned on page 14, line 308. Indicate what the letters stand for and who is included. 

Thank you for this reminder. We have added the following description to address this comment, lines 327 – 333:

Many advisors also noted challenges that students from backgrounds historically underrepresented in medicine (UIM) commonly faced, including lack of access to clinical volunteer opportunities, timely early advising, and multiple compounding barriers related to economic and familial pressures. Students considered to be UIM include those who identify as: Black/African American, Hispanic/Latino, Native American (American Indians, Alaska Natives, and Native Hawaiians), Pacific Islander, and mainland Puerto Rican as well as students from economically or educationally disadvantaged backgrounds.

3. Given that data from 3 different types of university systems are compared (i.e., CSU vs UC, vs private schools), the reviewers recommend that for each of the identified themes, examples from each of the university systems be included. This allows the reader to have a greater understanding of institutional variability or commonality. For example, for the theme “Competing Priorities for Advising” a reviewer noted that “there was not a discussion of institutional variation. And the quote provided from a CSU campus advisor is a phenomenon that may occur more frequently at public schools with first-generation and low-income students. “ 

We now include the following additional examples to highlight comparisons between university systems. 

Theme: Insufficient Institutional Investment in Advising

Added lines 229 - 232: One advisor from a private institution described his institution as “a business,” and explained that “ unless you can convince the administration that if you don’t put some money in the pre-health advising, you’re actually going to dimmish the way you look to your customer’s base. I don’t think they’re going to do it.” 

Theme: Competing priorities for advising

Added lines 252 - 254: Advisors at some UCs also noted institutional pressures to prioritize fulfillment of undergraduate major requirements over other interests, although such comments were less common. No private institutions noted institutional pressure related to increasing graduation rates.

Theme: Advisor turnover and training

Added lines 304 – 309: In contrast, one advisor at a private university highlighted their office’s ability to retain staff as a strength of their advising system. They explained: 

We have quite a large staff. We have a group where many of our staff have been with the office since its inception 11 years ago. I think we're really able to come together and find creative ways to support the students. I think that the staffing that we have allows us to do that. We also have a decent budget. – private school advisor 

Lines 323 – 325: On the other hand , schools with independent, career center or general academic advising structures less often cited lack of training as a challenge, with some even pointing toward their expertise in the field as an asset for students.

Theme: UIM Students Face Complex and Disproportionate Challenges 

Added lines 360 – 362: Advisors from all three university systems noted that UIM students at their campuses face disproportionate challenges, however, discussions of such challenges were more salient at the CSUs due to the high number of UIM students they serve. 

4. It is agreed that Table 3 provides an excellent example of potential innovations in pre-health advising. Reviewer 1 cautions it may be useful to note potential limitations for some of the recommendations. For example, peer advisors who serve as near-peer advisors sitting in for trained staff may provide inadequate or incorrect information to pre-health advisees. It is useful to note in this table, as was done for the “Committee letter”.

Thank you for this comment. We have added in the following limitations to Table 3: 

Peer advisors:

Although peer advisors can be a good resource, they should not take the place of professional advisors and should be supervised closely due to their lack of experience and training in the field. 

Connections with local physicians: 

Advisors noted that shadowing opportunities have been limited since COVID-19 restrictions were put in place.

Student club partnerships: 

Advisors noted that rapid turnover of club leadership poses a challenge to continuity of partnerships.

5. Overall, the topic is very interesting and has the potential to address factors associated with advising and racial/ethnic disparities in the health workforce. You are commended for your attempt to add to the extant literature on the topic of advising pre-health students. While the study focuses exclusively on pre-health students in California, it has implications for advising for all students attending institutions of higher education. 

Thank you very much for this comment, we appreciate your acknowledgement of the importance of this topic and hope this study will have implications for advising nationally. 

Reviewer #1

1. This study provides a concrete review of pre-health advising disparities in the state of California. CA is a useful case study because the implementation of Prop 209 in 1996 effectively did away from affirmative action for racially marginalized students. In the wake of the Supreme Court decision in Students for Fair Admissions, Inc. v. Harvard College, there are strong reasons to anticipate that pre-health students of color around the country will attend less selective institutions than they might have otherwise. The study attempts to gauge what kind of resources these students will find at less selective institutions. Resources matter for successful movement into health disciplines. Indeed, we see in CA huge differences between the racial background of state residents and representation of Latinx health providers. 

Thank you very much for this comment, we appreciate your acknowledgement of the importance of this topic and hope this study will have implications for advising nationally, especially given the recent SCOTUS decision. 

2. I am grateful to see clear information and comparisons across sectors in the state and find this article to be quite helpful in documenting disparities. The framing for why the problem is worthy of study is effective. 

Thank you for this comment and for letting us know that the framing of the problem was effective. 

3. Define UIM before using the term. Indicate what the letters stand for and who is included. 

Please see response to editor’s comments. 

4. Scholars frequently say they are doing grounded theory when they are not. Did the authors really do line-by-line coding with no presumptions of what they might find? My guess is that this article utilized more movement between deductive and inductive strategies (abductive analysis, Tavory and Timmermans 2014). Please provide a bit more clarity on the analysis technique and a citation to the closest methodological approach. 

Please see response to editor’s comments. 

5. Within each point, the authors need to give a sense of how common this was at CSU, vs. UC, vs. privates. They often do this well. However, on the “Competing Priorities for Advising” there was not a discussion of institutional variation. And the particular quote provided from a CSU campus advisor is a phenomenon that occurs, I would assume, much more frequently at public schools with first-gen and low-income students. Please contextualize. 

Please see response to editor’s comments. 

6. I love the table of potential innovations but note caution with peer advisors. I have also see cases where peer advisors sitting in for trained staff provide inadequate or incorrect information and are a poor band-aid for limited pre-health advising. It is useful to note in this table potential limitations of some of these remedies (as is done for the committee letter). 

Please see response to editor’s comments.

---

## [Editor Report · Decision Letter 1]

19 Dec 2023

Disparities in pre-health advising across California’s public universities

PONE-D-23-24270R1

Dear Dr. Rios-Fetchko,

We’re pleased to inform you that your manuscript has been judged scientifically suitable for publication and will be formally accepted for publication once it meets all outstanding technical requirements.

Kind regards,

Jocelyn Octavia Turner-Musa

Academic Editor

PLOS ONE

Additional Editor Comments (optional):

Thank you for resubmitting your manuscript with the recommended revisions. Once again, the authors are commended for addressing the topic of factors associated with advising and racial/ethnic disparities in the health workforce. The revised manuscript addresses concerns raised in the previous review

and is accepted.
---

## [Editor Report · Acceptance letter]

2 Jan 2024

PONE-D-23-24270R1 

PLOS ONE

Dear Dr. Rios-Fetchko, 

I'm pleased to inform you that your manuscript has been deemed suitable for publication in PLOS ONE. Congratulations! Your manuscript is now being handed over to our production team.

Kind regards, 

on behalf of

Dr. Jocelyn Octavia Turner-Musa 

Academic Editor

PLOS ONE